# Antioxidant, Immunomodulatory and Potential Anticancer Capacity of Polysaccharides (Glucans) from *Euglena gracilis G.A. Klebs*

**DOI:** 10.3390/ph15111379

**Published:** 2022-11-10

**Authors:** Virginia Casas-Arrojo, María de los Ángeles Arrojo Agudo, Casimiro Cárdenas García, Paloma Carrillo, Claudia Pérez Manríquez, Eduardo Martínez-Manzanares, Roberto T. Abdala Díaz

**Affiliations:** 1Departamento de Ecología, Facultad de Ciencias, Campus de Teatinos s/n, Universidad de Málaga, 29071 Málaga, Spain; 2Servicios Centrales de Apoyo a la Investigación (SCAI), Campus de Teatinos s/n, Universidad de Málaga, 29071 Málaga, Spain; 3Departamento de Biología Molecular y Bioquímica, Facultad de Ciencias, Campus de Teatinos s/n, Universidad de Málaga, 29071 Málaga, Spain; 4IBIMA (Biomedical Research Institute of Málaga), 29590 Málaga, Spain; 5Departamento de Botánica, Facultad de Ciencias Naturales y Oceanográficas, Universidad de Concepción, Concepción 4030000, Chile; 6Unidad de Desarrollo Tecnológico, Universidad de Concepción, Concepción 4190000, Chile; 7Departamento de Microbiología, Facultad de Medicina, Campus de Teatinos s/n, Universidad de Málaga, 29071 Málaga, Spain

**Keywords:** *Euglena gracilis*, polysaccharides, immunomodulatory capacity, antioxidant activity, anticancer capacity and nutraceuticals

## Abstract

The present study was carried out to determine the bioactivity of polysaccharides extracted from *Euglena gracilis* (EgPs). These were characterized by FT-IR and GC-MS. Cytotoxicity analyses (MTT) were performed on healthy human gingival fibroblast cell lines (HGF-1), obtaining an IC_50_ of 228.66 µg mL^−1^, and cell lines with anticancer activity for colon cancer (HCT-116), breast cancer (MCF-7), human leukemia (U-937, HL-60) and lung cancer (NCl-H460), showing that EgPs have anticancer activity, mainly in HTC-116 cells (IC_50_ = 26.1 µg mL^−1^). The immunological assay determined the immunomodulatory capacity of polysaccharides for the production of proinflammatory cytokines IL-6 and TNF-α in murine macrophages (RAW 264.7) and TNF-α in human monocytes (THP-1). It was observed that the EgPs had a stimulating capacity in the synthesis of these interleukins. The antioxidant capacity of polysaccharides and their biomass were analyzed using the ABTS method (18.30 ± 0.14% and (5.40 ± 0.56%, respectively, and the DPPH method for biomass (17.79 ± 0.57%). We quantitatively profiled HGF-1 proteins by liquid chromatography–tandem mass spectrometry analysis, coupled with 2-plex tandem mass tag labelling, in normal cells. In total, 1346 proteins were identified and quantified with high confidence, of which five were considered to be overexpressed. The data is available through ProteomeXchange, under identifier PXD029076.

## 1. Introduction

Microalgae are unicellular organisms classified as prokaryotes, and eukaryotes can grow autotrophic, heterotrophic, and some of the mixotrophic, modes. They are generally highly efficient at CO_2_ fixation and utilization of solar energy to produce biomass, with an efficiency up to four times higher than that of plants. The importance of microalgae lies in their role as primary producers in the food chain, in which they are the initial producers of organic matter.

These phytoplankton organisms have developed different strategies for survival. They are present in all environments with water, such as lakes, seas, and rivers, but can be found in soil and most terrestrial habitats, even the most extreme. They are widely distributed in the biosphere, are adapted to a lot of conditions, and constitute the largest group of living organisms, in terms of diversity of species in terrestrial and marine waters [1]. With more than 30,000 species of microalgae, only about 100 have been studied and ten are commercially relevant. Their ability to grow on a large scale, coupled with high efficiency in the production of organic compounds and rapid asexual reproduction by cell division, makes them attractive for use in food, cosmetics, pharmaceuticals, and biodiesel production [2]. In this sense, there are thousands of microalgae-based products ranging from nutraceuticals [2], feed for aquaculture fish or dietary supplements for animals, and cosmetics, such as creams, shampoos, etc. Nutraceuticals and nutritional supplements are a huge market for their multiple benefits [2].

Microalgae are capable of changing their metabolism according to the environmental conditions of the environment in which they are found, synthesizing various bioactive molecules, since they are rich in proteins, polysaccharides, pigments and fatty sulphates [2]. These biomolecules have a large number of biological activities, ranging from antimicrobial to antiviral, antitumor and even immunosuppressive activity, which are the main objectives in current biomedical research, ensuring the relevance of these biomolecules to research [3]. The importance of various physiological activities of algae polysaccharides, such as an anticoagulant, antiviral, antihyperlipidemic, and anticarcinogenic activities, are emphasized [4,5,6].

According to the system and culture conditions, the composition of microalgae (lipid content, carbohydrates, and proteins) varies, depending on the species considered, and within the same species.

*E. gracilis* is a eukaryotic microalga that does not have a cell wall, which makes it easily digestible. In addition, it stores large amounts of important nutrients, such as vitamins and minerals, that are easily absorbed. This, together with its easy cultivation, since it can grow as a strict photoautotroph, photoheterotroph or heterotroph using organic carbon sources [7], makes it one of the most studied eukaryotic microalgae, both at a biochemical and cell biology level.

*E. gracilis*, commonly known in Japan as “Midorimushi”, has been extensively studied as a producer of important molecules, such as vitamins A, C and E [8,9,10], as well as in the synthesizing of other biomolecules that are becoming increasingly important, such as essential amino acid sulphated polyunsaturated fatty acids [11], β-carotenes [8] and a glucan, paramylon, which is a high molecular weight unbranched β-(1,3)-D-glucan storage polysaccharide with immunomodulatory properties [12,13,14].

Paramylon is gaining importance for its potential medical and veterinary uses, although it does not yet have industrial use. This polysaccharide is a biologicall active. Other such polysaccharides include pachyman, lentinan, schizophyllan or fungal glucans, with important bioactivity, lentinan and schizophilic, which are tumor suppressors, and fungal glucans, which are well-known immunostimulants [15]. Paramylon has many useful applications in medicine as an immunostimulant and immunopotentiator. It has been shown to provide a remarkable range of health benefits, with stimulating effects on the immune system, moderating of blood glucose and insulin response, modest antitumor activity and cholesterol-lowering effects [16]. In addition, its sulphated derivatives have anti-HIV activity [17]. To produce paramylon efficiently, *E. gracilis* can be grown on potato liquor-based media in a heterotrophic mode. Some studies indicate that a repeated batch culture technique has the greatest potential for use in the large-scale production of paramylon by *E. gracilis* [18,19].

Due to their ease of culture and anticancer and immunomodulatory activities, this study determined the following objectives: to view the bioactivity of EgPs through *in-vitro* cytotoxicity (MTT) assays, particularly the antioxidant activity and immunomodulatory activity, and to identify and quantify the overexpressed proteins. To deepen understanding of the molecular mechanisms involved in the effects that the EgPs have on non-tumor cells, we investigated the proteins from HGF-1 cell extracts using a comparative proteomic approach by means of liquid chromatography high-resolution mass spectrometry analysis (LC/MS), coupled with 2-plex tandem mass tag labelling (TMT 2-plex). This research facilitated an in-depth study of EgPs and the discovery of metabolites of interest as a source of bioinspiration.

## 2. Results

### 2.1. Chemical Assessment

#### 2.1.1. Total Carbon (TC), Total Nitrogen (TN), Total Hydrogen (TH) and Total Sulphur (TS)

In the present study of *E. gracilis* elemental analysis was performed on the algal biomass and the extracted polysaccharides. Biomass showed a high percentage of carbon (44.59%), followed by nitrogen (10.54%) and hydrogen (5.57%). The sulphur content was practically null (0.03%).

Concerning the elemental analysis of the EgPs, the highest content was C, with 1.24%, followed by N and H, with 0.24 and 0.20%, respectively. The S content was also analyzed, obtaining a higher percentage of Sin polysaccharides than in biomass with 0.16% (Table 1).

Accordingly, the obtained molar (C/N ratio) value was 4.93 in biomass and 5.27 in polysaccharides.

#### 2.1.2. Protein, Carbohydrates, Lipids, Inorganic Compounds and Moisture

Analysis of *E. gracilis* biomass is summarized in Table 2, i.e., total proteins, carbohydrates, lipids, moisture, and inorganic compounds.

#### 2.1.3. Fourier Transform Infrared Spectroscopy (FT-IR)

FT-IR spectrum was recorded in the region of 4000–400 cm^−1^. Figure 1 shows the FT-IR spectrum of EgPs, which exhibited the characteristic band features for polysaccharide fractions. EgPs showed the broadband positioned at 3351 cm^−1^, which was a distinctive sign of the band of axial stretching of OH. The band at 2954 cm^−1^ and 2919 cm^−1^ signified the presence of C-H groups (Str of asymmetric (CH_2_) and Str of aliphatic (C–H)). The signals around 1640 cm^−1^ and 1405 cm^−1^ were attributed to asymmetric and symmetric stretching of a carboxylate anion group (C=O). The stretching at 1226 cm^−1^ identified the S=O vibration and showed the presence of the sulphate group. The intense absorption band at around 1068 cm^−1^ was attributed to the coupling of C-O-C, C-C stretching mode, and C-O-H bending mode in compounds like sugar. The FT-IR spectrum of EgPs exhibited the signal at 908 cm^−1^ which was attributed to typical β-configurations, indicating the presence of β-glucans. Additionally, the signal at 882 cm^−1^ represented the presence of β-pyranoses. This result confirmed the presence of sulphate groups in EgPs.

#### 2.1.4. Gas Chromatography–Mass Spectrometry (GC-MS)

The major peak in the GC–MS spectrum of polysaccharides corresponded to the monosaccharide glucose, with a retention time of 29.73 min (Figure 2; Table 3). The other peaks belonged to ribose and mannose. Other monosaccharides with lower relative abundances were identified as galactose and fucose.

### 2.2. Biological Assessment

#### 2.2.1. Antioxidant Activity (ABTS Method)

The results showed a maximum value of antioxidant activity for the polysaccharides and the biomass of *E. gracilis* of 5.40 ± 0.26% and 18.30 ± 0.14%, respectively, at the concentration of 500 µg mL^−1^ (Figure 3).

#### 2.2.2. Antioxidant Activity (DPPH Method)

The antioxidant activity assay, using the DPPH method, for the biomass from *E. gracilis* showed the highest antioxidant activity at the concentration of 200 µg mL^−1^ with a value of 17.79 ± 0.57%, although at the studied concentrations, there was not much variation between the data obtained from this method. (Figure 4).

#### 2.2.3. Cell Viability of Lines HTC-116, MCF-7, U-937, HL-60 and NCl-H460

The HTC-116, MCF-7, U-937, HL-60, and NCl-H460 cells were treated with different concentrations of EgPs to assay the potential effects of this compound on anticancer activity. The results obtained with the MTT assay showed that the tested polysaccharides inhibited cell proliferation in a dose-dependent manner, so its anticancer ability in these tumor cells was shown.

According to results obtained from the anticancer activity assay of EgPs, it was observed that the highest IC_50_ value was obtained in MCF-7 (110.53 µg mL^−1^) (Figure 5C), followed by NCl-H460 (107.11 µg mL^−1^) (Figure 5B). In the case of HTC-116, the IC_50_ was lower than in the others, being 26.10 µg mL^−1^ (Figure 5A). In the U-937 and HL-60 cells, IC_50_ of 27.54 µg mL^−1^ was obtained (Figure 5D) and 18.20 µg mL^−1^ (Figure 5E) respectively. These values indicated that these polysaccharides have great potential against these cells at a very low level to inhibit cell growth in concentrations of 50%.

#### 2.2.4. Cell Viability of Line HFG-1 more and less Proliferating

This test was carried out to make a comparative study between healthy cells (HGF-1) in a proliferative state, and, therefore, simulating the typical state of tumor cells, and in a healthy state.

These cells were treated with different concentrations of EgPs to test the potential effects of this compound in both conditions (Figure 6 and Figure 7). The results obtained with the MTT assays showed that the tested polysaccharides inhibited cell proliferation in a dose-dependent manner, obtaining an IC_50_ value one order of magnitude lower in proliferative HGF-1 (similar to cancerous conditions) cells, compared to HGF-1 less proliferative (normal healthy condition) cells. That is, the IC_50_ value in more proliferating HGF-1 cells with EgPs was 147.86 µg mL^−1^ (Figure 6), while the IC_50_ value of less proliferating HGF-1 cells was 228.66 µg mL^−1^ (Figure 7).

#### 2.2.5. Determination of Cytokines (IL-6 and TNF-α) with RAW 264.7 Cell Line

This trial evaluated the production of proinflammatory cytokines, TNF-α and IL-6. For this purpose, the RAW 264.7 mouse macrophage cell line was used with the following polysaccharide concentrations: 1 μg mL^−1^, 5 μg mL^−1^, 25 μg mL^−1^, 50 μg mL^−1^, 75 μg mL^−1,^ and 100 μg mL^−1^. For the positive control, *E. coli* lipopolysaccharide (LPS) was used at a concentration of 0.5 μg mL^−1^.

The maximum concentration of Interleukin-6 obtained within the concentrations studied was 3056.60 pg mL^−1^ when a concentration of polysaccharides of 100 μg mL^−1^ (Figure 8) was applied in the case of TNF-α, and was 564.05 pg mL^−1^ (Figure 9). Both figures show that for IL-6 and TNF-α, the synthesis and accumulation by RAW 264.7 cells increased as a higher concentration of EgPs was applied. Specifically, IL-6 synthesis increased exponentially from 108 to 3057 pg mL^−1^ as it passed a polysaccharide concentration of 5 to 100 μg mL^−1^. Similarly, TNF-α production increased exponentially from 87 to 564 pg mL^−1^ when the polysaccharide concentration increased from 5 to 100 μg mL^−1^. However, in none of the cases did the cytokine levels become saturated.

#### 2.2.6. Determination of TNF-α with Human THP-1 Cell Line

Human THP-1 were exposed to 10 ng mL^−1^ phorbol-12-myristate-13-acetate (PMA) for 48 h to differentiate them into macrophages and to ensure that samples with detectable quantities of TNF-α were obtained. After that, THP-1-derived macrophages were exposed to different concentrations of EgPs for 24 h. As shown in Figure 10, cells exposed to crescent concentrations of EgPs increased TNF-α production from a concentration of 15.6 µg mL^−1^, reaching its maximum effect at 250 µg mL^−1^.

#### 2.2.7. Proteomic Analysis in HGF-1 Cells

We quantified the proteomic changes between treated and control HGF-1 cell groups utilizing TMT isobaric labelling of the protein extracts derived from culture cells, followed by ultra-high-performance liquid chromatography–high-resolution mass spectrometry (UHPLC-HRMS). Data analysis identified a total of 1346 proteins with high confidence (<1% false discovery rate and more than two different peptides identified per protein). In total, five proteins were upregulated (Log_2_ fold-change > 0.3, *p*-value < 0.05) as compared with the controls. Significantly changed proteins are listed in Table 4 and illustrated in Figure 11.

### 2.3. STRING Analysis of Protein Networks

In addition, we carried out a bioinformatics analysis of protein interaction networks for upregulated proteins, which was built using the database and web-tool STRING (Search Tool for the Retrieval of Interacting Genes/Proteins) version 11.5 [20]. This analysis summarized the network of predicted associations for TNF-α, IL-6, and the upregulated proteins in HGF-1 cells after treatment with EgPs (Figure 12)*,* allowing us to delve deeper into the functional pathways where these proteins were involved. The analysis of the networks showed that proteins had more interactions among themselves than what would be expected for a random set of proteins of the same size and degree distribution drawn from the genome (11 vs. 5), as shown in Figure 12C, which meant that the proteins were at least partially biologically connected, as a group. Therefore TNF-α and IL-6 were significantly linked to the overexpressed proteins. Table 5 reports all the significant interactions between the different proteins.

## 3. Discussion

The total content of C and N in the biomass *of E. gracilis* was higher than that obtained in other microalgae, such as *Tetraselmis suecica* [21] or *Porphyridium purpureum* [22], as was the content of H, with respect to *P. purpureum*. However, the S content was lower than that of *P. purpureum*. Regarding the polysaccharides, it was observed that the C content of the EgPs was lower than that obtained in the polysaccharides of *T. suecica* [21] and *P. purpureum* [22]. In the case of total Nitrogen, it was lower than that of *T. suecica*, and higher than in *P. purpureum*. The H content was lower in EgPs than in the polysaccharides *of P. purpureum*. When comparing the content of S, it was observed that the same percentage was obtained in EgPs and in polysaccharides of *P. purpureum.*

The study by Metsoviti et al. 2019 [23] compared the growth rates and nutrient contents of five species of microalgae grown in greenhouses, including *E. gracilis*, the samples of which were taken under different conditions of light intensity and temperature. The data for ash, protein and moisture content were lower than those obtained in our study in all cases, while the percentage of lipids varied according to light intensity and temperature, with the lipid content being higher than that obtained in our study at a higher intensity of light and temperature, and lower in conditions of lower intensity of light and temperature. If we compare our results with different strains of *Tetraselmis* and with *Pyramimonas* from the study by Tzovenis et al. 2009 [24], the contents of inorganic compounds, and mainly carbohydrates, were higher than that obtained with *E. gracilis*. However, the percentages of proteins and lipids were lower than that obtained in *E. gracilis*. It is known that the concentration of lipids in microalgae is extremely variable, reaching productive peaks of 70–80% for those used in oil production [25].

The bands obtained in the FT–IR of the EgPs coincided with those obtained in polysaccharides from other algae, as determined by different authors. In the case of the band at 3351 cm^−1^, this was also found in polysaccharides from *Sarcodia ceylonensis, Ulva lactuca, Gracilaria lemaneiformis* and *Durvillaea antarctica*, according to He et al. (2016) [26]. An important reserve polysaccharide in Euglenophytes is the paramylon that presents bands at 2954 cm^−1^ and 2919 cm^−1^ corresponding to the CH group, as indicated by Synytsya, A. & Novak, M (2014) [27], and a peak at 908 cm^−1^, characteristic of β-glucans [28], and at 882 cm^−1^ of β-pyranose [26]. The presence of sulphate groups (S=O) and carboxylate anions (C=O) identified in EgPs have also been described by other authors in polysaccharides from algae [29,30,31]. The appearance of intense bands around 1068 cm^−1^ was attributed to the presence of compounds, such as sugars [32,33]. According to Meng et al. (2014) [34], the FTIR method is a validated spectroscopic method, which determines the existence of polysaccharides, lipids and proteins in algal samples and evaluates the physiology of microalgae. This study indicated that the bands at 1200–950 cm^−1^ characterized the absorption of C-O-C vibrations of polysaccharides, as was detected in our samples. This characterization method could be used to temporarily observe the extraction dynamics of different algae compounds and compare them with spectra from digital libraries, which would allow the identification of fractions of biotechnological interest and industrial application.

The identification of the derivatized samples was carried out in a GC–MS system, particularly by comparing their retention times with those of the standard monosaccharides Arb, Rib, Fuc, Glc, Gal, Rha, Fru, Mann, Xyl, Gluc-A and Api. The monosaccharide units identified were mainly Glc, although Rib and Mann, and other minor monosaccharides, such as Gal and Fuc, were also detected. There have been very few characterization tests of EgPs conducted. However, those that have been conducted agree with the results obtained in our test. Thus, Barras and Stone, 1965 [35] indicated that Glc, Gal, Mann, Fuc, Xyl and Rha were detected on the cell surface of *E. gracilis*, as well as Xyl, associated with flagella [36]. Farrah et al., 2019 [37] detected glucose and matinol in the aqueous extracts of *E. gracilis* in GC–MS, this being a derivative of mannose. The differences between the carbohydrate composition of *E. gracilis* in our and other studies could be explained by differences in the extraction methods, the type of derivation, and the carbohydrate analysis strategies.

There are studies that show the antioxidant activity of microalgae [38,39]. These contain various compounds that have antioxidant characteristics, such as carotenoids, vitamins C and E, florotannines, ω3 polyunsaturated fatty sulphates, polysaccharides and others [40]. However, the most interesting ones are carotenoids, especially astaxanthin, β-carotene and lutein [41]. According to [42], the antioxidant activity of the biomass, determined by means of the ABTS method in *Galdieria sulphuraria*, has much higher activity than that obtained at the studied concentrations of *E. gracilis* and the same happens with *Spirulina plantensis* [43]. However, the activity of *E. gracilis* is comparable and, in many cases, superior, to various foods, such as extracts of fruits, vegetables, and drinks, as demonstrated in the study carried out by Pellegrini et al., 2003 [44]. In the case of the biomass test using the DPPH method, the antioxidant activity of *S. platensis* was higher than that obtained in *E. gracilis* [45]. The antioxidant activity of EgPs, determined by means of the ABTS test was lower than the polysaccharides of other microalgae, such as *Porphyridium cruentum* [46] as well as the *S. plantensis* [47] polysaccharides. It was shown that *Tetraselmis suecica* polysaccharides have antioxidant activity in both autotrophic and heterotrophic cultures in [21].This indicated that the antioxidant activity was higher or lower according to different factors, such as the monomers that form the polysaccharides [48], the oligomeric components, sulphates and glycoproteins of the polymer or the combination of these factors [49]. Other authors, [50,51] reported similar values of antioxidant activity (35–40 and 66.5 µmol Trolox g^−1^ DW) in polysaccharides from the fungus *Cordyceps sinensis*.

Studies show that algae (micro and macroalgae) produce polysaccharides with biological activities: anti-inflammatory, antioxidant, antiviral, anticoagulant, anticancer and immunomodulant [52]. The results obtained in this research showed anticancer activity of EgPs in the cell lines studied (U-937, HL-60, NCl-H460, HTC-116 and MCF-7). Microalgae polysaccharides are interesting candidates for antitumor therapies. Polysaccharides from *Tribonema* sp. and *Phaedactylum tricornum* induced apoptosis in the liver cancer cell line (HepG2) [53,54]. The anticancer activity of polysaccharides from *S. platensis*, on the cell lines MCF-7 and HepG2 (liver hepatocellular carcinoma), were studied by Abd El Baky H. et al. (2016), whose IC_50_ results were 81.85 and 80.59 µg mL^−1^, respectively [47]. Ket et al. (2003) [55] demonstrated the anticancer capacity of polysaccharides from *Gymnodinium sp. A3 (GA3)* on a large number of tumor lines, obtaining very low growth inhibition (IC_50_) results. For the cell lines HTC-116 and MCF-7 they obtained IC_50_ of 3.7 and 2.9 µg mL^−1^, respectively. Polysaccharides of *T. suecica* also exhibited anticancer activity, mainly in the HL-60 cell line, presenting an IC_50_ of 36 μg mL^−1^ [21].

Polysaccharides from various algae [56,57] and higher plants [58,59] have been reported to increase phagocytic and phagocyte-secreting activities. According to the data obtained, the effect of EgPs on the immune system is due to a strong stimulation of the macrophages in relation to the synthesis of interleukins with a maximum concentration in Il-6 of 3056.60 pg mL^−1^ and in TNF-α of 564.05 pg mL^−1^. As is known, polysaccharides from numerous microalgae have been reported to stimulate the activity of macrophages and phagocytes. For example, according to [60] the polysaccharides extracted from *P. cruentum* are potent inducers of cytokine secretion IL-6 and TNF-α [22]. De Jesus et al. (2013) [52] indicated that polysaccharides of marine microalgae, such as *Porphyridium, Phaeodactylum* and *Chlorella stigmatophora*, showed anti-inflammatory activity and acted as immunomodulatory agents. The direct stimulating effect of *Phaeodactylum tricornutum* on immune cells was evidenced by positive phagocytic activity assayed in vitro or in vivo. Polysaccharides from other algae, such as *Laminaria ochroleuca, Porphyra umbilicalis* and *Gelidium corneum* [61], have also been studied for their immunomodulatory ability, demonstrating the high capacity of these molecules to activate macrophages and for secretion of pro-inflammatory cytokines IL-6 and TNF-α, especially the polysaccharides from *Laminaria ochroleuca*. Sun [62] showed that EPS (sulphated polysaccharides) showed immunostimulant activity in mice with S180 tumors, by increasing both the spleen index and thymus index, and also the spleen lymphocyte index. It has also been reported that polysaccharides from *P. cruentum* induced an increase in the production of TNF-α and IL-6 in the mouse macrophage cell line (RAW 264.7) [22]. These results demonstrate the ability of algae polysaccharides to directly stimulate the immune system.

In the present study, we analyzed the proteomic profile of control and EgP-treated HGF-1 cells to understand the molecular mechanisms underlying their biological effects. A comparative proteomics approach allowed us to identify five key proteins that could be involved in their biological activity: alpha-2-HS-glycoprotein (AHSG), myosin regulatory light chain 12B (MYL12B), thymosin beta-10 (TMSB10), Superoxide dismutase-2 (SOD2) and metallothionein-2 (MT2). Four of them have been reported as anti-inflammatory agents: AHSG, MYL12B, SOD2, and MT2. AHSG deficiency is associated with inflammation and links vascular calcification to mortality in patients on dialysis [63]. Although MYL12B has its most important role in regulation of cytokinesis and cell locomotion [64], it was recently reported as a ligand of the CD69 molecule, a type II transmembrane protein that is recognized as a marker of lymphocyte activation [65]. In addition, many studies have shown that there are antioxidant enzymes that have protective effects during inflammation by free radical scavenging, as occurs with SOD. These play an essential role in inflammatory diseases because they catalyze the conversion of superoxide into hydrogen peroxide and oxygen, affecting immune responses [66]. As is the case with SOD, MT2 also acts as an antioxidant. Mammals have four general metallothionein isoforms (MT1, 2, 3, 4). Though the functions of metallothioneins (MTs) have not been fully elucidated, they appear to participate in detoxifying heavy metals [67], storing and transporting zinc, and redox biochemistry [68]. In fact, the redox regulating capacity of MTs is approximately 50 times greater than that of the major antioxidant, glutathione [69]. It has been suggested that induction of MTs in cell therapy may provide protection by serving as antioxidant and anti-inflammatory agents [70]. What is more, cytokines, such as TNF-α and IL-6 modulate MTs and Zn metabolism in non-immunological organs, such as the hepatic tissue [71].

Regarding the antitumor effects that EgPs could have, several observations suggest that TMSB10 plays a significant. and possibly obligatory, role in cellular processes controlling apoptosis, possibly by acting as an actin-mediated tumor suppressor [72].

## 4. Materials and Methods

### 4.1. Biological Material

*E. gracilis* used for the realization of this study was within the species that we had in our cepari, in this case the coding was Eg 01. Once collected from the cepari, *E. gracilis* was inoculated into a 1 litre flask for growth. The medium used was 3N-BBM + V (Bold Basal Medium with 3 times Nitrogen and Vitamins, modified) enriched with vitamin B12 (Sigma-Aldrich, St. Louis, MO, USA), B1 (Sigma-Aldrich, St. Louis, MO, USA) and Biotin (Sigma-Aldrich, St. Louis, MO, USA). The culture was kept at a pH 7 in a culture chamber at 25 °C, 100 μmol of photons m^−2^ s^−1^ lighting and in constant agitation by air bubbling. Once a steady state was reached, the biomass was inoculated in larger flasks, maintaining the same conditions, to obtain a greater amount of biomass in order to carry out all the subsequent tests.

### 4.2. Total Carbon (C), Hydrogen (H), Nitrogen (N) and Sulphur (S)

To determine the total carbon, nitrogen and sulphur of the biomass and polysaccharide samples, the combustion technique was used, using the LECO TruSpec Micro CHNSO elemental analyser (St. Joseph, MI, USA). The result obtained was given in % with respect to the weight of the sample for each of the elements (C, H, N, S).

### 4.3. Protein Content

Once the samples were analysed with the CNH Perkin-Elmer 2400 Mar Biotechnol (Perkin-Elmer, Waltham, MA, USA) analyzer, the protein content was calculated by multiplying the per cent of total nitrogen by a factor reported by Lourenço 4.99 [73]. To calculate the total protein concentration from the nitrogen content in the microalgae in the early stationary phase, the use of the average overall N-Prot Factor of 4.99 is recommended, as it has not yet been studied for its specific factor.

### 4.4. Determination of Total Carbohydrates

The total carbohydrates were determined according to the method of Dubois [74]. The technique is described by Kochert [75]. Five mg of lyophilized biomass of *E. gracilis* were taken and 1 mL of H_2_SO_4_ (1 M) was added. The mixture was sonicated for 5 min. Then, 4 mL of H_2_SO_4_ (1 M) (Sigma Aldrich, St. Louis, MO, USA) were added and the mixture was placed in a water bath (temperature 100 °C) for 1 h. After that, the samples were cooled to ambient temperature. Then, the samples were centrifuged at 4000 rpm at 10 °C for 15 min. After this, 0.25 mL of the samples were taken. This extract was mixed with 0.75 mL of H_2_SO_4_ (1 M) and 1 mL of 5% phenol (Sigma Aldrich, St. Louis, MO, USA) and allowed to stand for 40 min at room temperature. Finally, 5 mL of concentrated H_2_SO_4_ was slowly added. After cooling to room temperature, the sample was measured using a spectrophotometer at 485 nm. The blank sample was made in the same way, substituting the sulphated extract of the sample with 1 mL of H_2_SO_4_ (1 M).

### 4.5. Determination of Lipids

Lipids were determined by the Folch method [76]. For this purpose, 200 mg of lyophilized algae were taken and 5 mL of chloroform (Sigma Aldrich, St. Louis, MO, USA) and methanol mixture (2:1) was added with 0.01% butylhydroxytoluene (BHT) (Sigma Aldrich St. Louis, MO, USA). The solution was homogenized, and 2 mL of 0.88% KCl was added. The mixture was centrifuged for 5 min at 2000 rpm. Two phases appeared. The upper phase was discarded, and the lower was collected and filtered. It was cooled at −20 °C for at least 20 min. The thin layer of salt that formed on the surface was removed. The lipids were weighed and stored in a nitrogen atmosphere.

The percentage (%) of lipids was calculated according to the Equation (1):Lipids (%) = [Lipids (g)/Biomass (g DW)] × 100(1)

### 4.6. Determination of Inorganic Compounds

To determine the inorganic compounds, a weight of 2 g of DW of *E. gracilis* was placed into a previously weighed crucible. The crucible was placed in a muffle for 24 h at 550 °C. During this process, all organic matter was eliminated. The weight of the crucible with the samples was determined by subtracting the final weight of the crucible from the initial weight. Finally, the percentage of inorganic compounds in the sample was calculated.

### 4.7. Extraction of Polysaccharides

The polysaccharide extraction process was carried out according to Pajares et al. (2012) [77] with some modifications. For this extraction, we first depigmented the biomass, for which 10 g of dry biomass of *E. gracilis* were taken, added in 400 mL of EtOH (Sigma-Aldrich, St. Louis, MO, USA) at 80% (*v*/*v*), stirred at room temperature for 16 h, and then centrifuged (4000 rpm, 10 min). This was done repeatedly until a colourless supernatant was obtained (usually after four times). After the last centrifugation, the pellet was taken and added to 300 mL of dH_2_O, and heated at 100 °C with continuous stirring for 40 min. After this time, it was allowed to cool and centrifuged (4500 rpm, 15 min). The supernatant was saved for further processing. This was repeated up to two times so that the greatest possible quantity of polysaccharides was extracted. The supernatants obtained were mixed in a beaker and the alginates were removed with polyvinylpolypyrrolidone (PVPP) (Sigma-Aldrich, St. Louis, MO, USA) reagent. To do this, 2 spatula tips were added to the supernatant, it was stirred and left for 5 min, then centrifuged at 3000 rpm for 5 min, eliminating the polyphenols. Once this was done, the entire supernatant was poured back into a beaker and placed on a tray with ice, 2% (*w*/*v*) O-N-cetylpyridinium bromide (Cetavlon) (Merk, Darmstadt, Germany) was added according to Abdala Díaz et al. (2010) [78] for the selective precipitation of polysaccharides. The polysaccharides were flocculated after this process and were recovered by centrifugation (4500 rpm, 10 min). The precipitated polysaccharide was purified with 4 M NaCl (Sigma-Aldrich, St. Louis, MO, USA) and the solution was stirred at 60 °C for 30 min. The polysaccharide was flocculated again with 96% (*v*/*v*) ethanol (Sigma-Aldrich, St. Louis, MO, USA), centrifuged (4500 rpm, 10 min). The pellet was recovered and placed in a dialysis membrane (Dialysis Tubing Cellulose, Sigma Aldrich St. Louis, MO, USA).The membrane with the polysaccharides was placed in a 0.5 M NaCl solution and shaken overnight at 4 °C. After dialysis, the entire content of the membrane was recovered, and washed with 96% EtOH. The mixture was kept at 4 °C until the flocculation of the polysaccharides had occurred. A final centrifugation pulse (3000 rpm 2 min, room temperature) was applied to recover the polysaccharides, which were then frozen at −80 °C and lyophilized (Lyophilizer Cryodos, Telstar, Terrasa, Spain).

### 4.8. Fourier Transform Infrared Spectroscopy (FT-IR)

To perform the Fourier transform infrared characterization, the spectra were collected in a Thermo Nicolet Avatar 360 IR spectrophotometer (Thermo Electron Inc., Waltham, MA, USA) with a resolution of 4 cm^−1^, equipped with a DTGS detector and OMNIC software 7.2 (Thermo Nicolet, Waltham, MA, USA). Baseline adjustment was performed using Thermo Nicolet OMNIC (Thermo Nicolet, Waltham, MA, USA) software to flatten the baseline in each spectrum. For this, a self-supporting disc had been previously made, pressed from the mixture of EgPs-KBr (1% p/p), which was introduced in the spectrophotometer for its characterization.

### 4.9. Gas Chromatography–Mass Spectrometry (GC-MS)

#### 4.9.1. Derivatization of Polysaccharides

EgP samples (2 mg) and monosaccharide standards were treated with the same procedure. First, 100 µL of the standard stock solution of 1 mg mL^−1^ of each monosaccharide was dried under nitrogen gas flow. Second, the samples of polysaccharides, and a mixture containing the standard monosaccharides included in the Internal Standard (IS), were methanolyzed in 2 mL metanol/3 M HCl at 80 °C for 24 h. The monosaccharides, glucose, galactose, rhamnose, fructose, mannose, xylose, apiose, and myo-inositol (IS) (Sigma-Aldrich, St. Louis, MO, USA), as well as pyridine, hexane and metanol/3 M HCl solution were purchased from Sigma-Aldrich (St. Louis, MO, USA). Then, the saccharides were washed with methanol and dried under nitrogen gas flow. Third, the Trimethylsilyl reaction was accomplished with 200 µL of Tri-Sil HTP (Thermo Fisher Scientific, Franklin, MA, USA). Each vial with the sample was heated at 80 °C for 1 h. The derived sample was cooled to room temperature and dried under a stream of nitrogen. Fourth, the dry residue was extracted with hexane (2 mL) (Sigma-Aldrich, St. Louis, MO, USA), and centrifuged. Finally, the hexane solution containing silylated monosaccharides was concentrated and reconstituted in hexane (200 µL), filtered and transferred to a GC-MS autosampler vial (Thermo Fisher Scientific, Franklin, MA, USA). Sample preparation and analyses were performed in triplicate.

#### 4.9.2. Gas Chromatography/Mass Spectrometry (GC-MS) Analysis

GC/MS analyses were carried out using a gas chromatography Trace GC (Thermo Fisher Scientific, Franklin, MA, USA), autosampler Tri Plus and DSQ mass spectrometer quadrupole (Thermo Fisher Scientific, Franklin, MA, USA). The column was ZB-5 Zebron, Phenomenex (5% Phenyl, 95% Dimethylpolysiloxane) of dimensions 30 m × 0.25 mm i.d. ×0.25 µm. The column temperature was programmed from 80 °C (held 2 min) and 5 °C min^−1^ to a final temperature of 230 °C. The carrier gas was helium (flow 1.2 mL min^−1^). The injection volume was 1 µL in a splitless mode at 250 °C. The source and MS (Thermo Fisher Scientific, Franklin, MA, USA) transfer line temperature were 230 °C. The mass spectrometer was set for a Select Ion Monitoring (SIM) (Thermo Fisher Scientific, Franklin, MA, USA) program in electron ionization mode (EI) at 70 eV. The TMS-derivatives were identified by characteristic retention times and mass spectrum compared to those of authentic standards. The compounds were identified by comparing the mass spectra with those in the National Institute of Standards and Technology (NIST 2014) library.

### 4.10. Antioxidant Activity (ABTS Method) in Polysaccharides and Biomass

The ability of EgPs to scavenge the ABTS radicals was evaluated sing an ABTS assay, as reported by Re et al. (1999) [79], with a few modifications. ABTS radical cation was generated by a reaction of 7 mM ABTS (Sigma-Aldrich, St. Louis, MO, USA) with 2.45 mM potassium persulphate (Sigma-Aldrich, St. Louis, MO, USA). This reaction mixture was stored for 16 h at room temperature and used within two days. After incubation, the mixed solution was diluted to a 0.7 absorbance unit at 413 nm with the deionized water. From a weight of 10 mg of the lyophilized samples (biomass and polysaccharides) dissolved in 1 mL of phosphate, buffer dilutions were made so that the final concentrations in the cuvettes were 12.5, 25, 50, 100, and 200 µg mL^−1^. Then, 50 µL of these samples were mixed with 940 µL of phosphate buffer and 10 µL of ABTS solution in the cuvettes. Then, the mixture was determined at 413 nm. ABTS radical scavenging capacity was calculated according to the Equation (2):AA% = [Abs (control) − Abs (sample) Abs (control)] × 100(2)where, A_0_ is the absorbance of the ABTS radical in phosphate buffer at time 0; A_1_ is the absorbance of the ABTS radical solution mixed with the sample after 8 min. From a stock of Trolox^®^ 2.5 mM a calibration curve was made with different concentrations of Trolox^®^. The serial dilutions were obtained at concentrations of 5, 10, 15, and 20 μM. All determinations were performed in triplicate (*n* = 3) [80]. The antioxidant capacity was expressed as % antioxidant activity.

### 4.11. Antioxidant Activity (DPPH Method) in Biomass

The antioxidant and radical scavenging activities of biomass extracts were evaluated by the 2,2-diphenyl-1-picrylhydrazil (DPPH) free radical method of Brand-Williams et al., (1995) [81]. From 1 mg of dry weight of the crushed algae, the biomass extract was obtained, and 1 mL of MeOH at 80% was added. Dilutions were made so that the final concentrations in the cuvettes were 200, 100, 50, 25 and 12.5 µg mL^−1^. DPPH (Sigma-Aldrich, St. Louis, MO, USA) (1000 µL), with 80% MeOH and 200 µL sample, was added to the cuvettes. The initial absorbance was measured at 517 nm. The samples were incubated for 30 min and the absorbance was recorded again at 517 nm, using 80% MeOH (Sigma-Aldrich, St. Louis, MO, USA) as the target. The absorbance was then transformed into a percentage inhibition against 80% MeOH.

The percentage of the antioxidant activity was calculated according to Equation (3):AA% = [(Abs_0_ − Abs_1_) Abs_0_] × 100(3)where, A_0_ is the absorbance at time 0 min and A_1_ is the absorbance at the end of the reaction (30 min) at 517 nm. A calibration curve was made with different concentrations of Trolox^®^, from a stock of Trolox^®^ 1.268 mM. Dilutions were obtained at concentrations from 0 to 6 μM. All determinations were performed in triplicate (*n* = 3) [80]. The antioxidant capacity was expressed as % of antioxidant activity.

### 4.12. Lipopolysaccharides (LPS) Contamination Assay

LPS contamination in EgP was evaluated using the Limulus Endosafe^®^-PTS amebocyte lysate (LAL) assay kit (Charles River Laboratories, Charleston, SC, USA) following the manufacturer’s instructions. For this, 25 μL of the polysaccharide solution were loaded into each of the four channels of the cartridge. The LAL reagent and sample were automatically mixed by the reader in two channels, just like the sample, with the LAL reagent and positive product in the other two channels, as a control. The channels were incubated and combined with the chromogenic substrate. Following this, the optical densities of all four channels were measured and compared to an internal standard curve. The result of the amount of endotoxin in the sample was expressed as endotoxin units (EU) mL^−1^.

### 4.13. Cell Culture

In this study, six cell lines were used: human colon cancer cell line (HCT-116, ATCC, Manassas, VA, USA), breast adenocarcinoma human cell line (MCF-7, ATCC, Manassas, VA, USA), human leukemia cell line (U-937 and HL-60 ATCC, Manassas, VA, USA), lung cancer (NCl-H460, ATCC, Manassas, VA, USA), murine macrophages cell line (RAW 264.7, ATCC, Manassas, VA, USA), human gingival fibroblasts cell line (HGF-1, primary culture) and human monocytes cell line (THP-1, ATCC, Manassas, VA, USA).

HCT-116, MCF-7, RAW 264.7, and HGF-1 were routinely grown in Dulbecco’s Modified Eagle’s Medium (DMEM) (Biowest, Nuaillé, France), supplemented with 10% fetal bovine serum (FBS) (Biowest, Nuaillé, France), 2 mM L-glutamine (Biowest, Nuaillé, France), 5 mL penicillin-streptomycin (Biowest, Nuaillé, France) and 2.5 mL amphotericin B (Biowest, Nuaillé, France). The U-937 and HCl-N460 were cultured in RPMI-1640 medium (Biowest, Nuaillé, France), supplemented with 10% FBS, 2 mM L-glutamine, 5 mL penicillin-streptomycin, and 2.5 mL amphotericin B. In the case of HL-60, 5 mL of penicillin-streptomycin and 2.5 mL of amphotericin B and THP-1 were grown in RPMI-1640 medium, supplemented with 20% FBS, with 0.05 mM, 2-mercaptoethanol (Sigma-Aldrich, St. Luis, MO, USA), 10% FBS, 5 mL penicillin-streptomycin, and 2.5 mL amphotericin B. These were incubated at 37 °C in humidified air with 5% CO_2_. Cultured cells were harvested when confluency reached 80%.

### 4.14. MTT Assay in Tumoral Cell Lines

For the MTT assay different tumor cell lines (U-937, HL-60, HTC-116, NCl-H460 and MCF-7) were incubated at polysaccharide concentrations from 10,000 to 4.77 × 10^−3^ µg mL^−1^ in series. Then, 1:1 dilutions of EgPs from each cell line were placed in a 96-well microplate for 72 h at 37 °C, 5% CO_2_. The cellular proliferation of the lines used was estimated by the MTT assay (3-(4,5-dimethylthiazol-2-yl)-2,5-diphenyltetrazolium bromide) (Sigma-Aldrich, St. Luis, MO, USA) [78]. Determinations were carried out in independent experiments in triplicate.

### 4.15. Cytotoxicity Assay MTT with Healthy Cell Line (HGF-1)

MTT assays were performed with HGF-1 cells. The same supplements were used at the same concentrations except for FBS. To resemble what occurs in tumor cells, less cells (very proliferative) were taken and the concentration of FBS used was 10%. In the case of comparing healthy cell behaviour, the number of cells taken was ten times more than the amount previously taken (low proliferating), and the FBS concentration was 2%. With this test, we wanted to observe the behaviour of the cells in different amounts against the treatment used in our study.

HGF-1 cells were incubated with different concentrations of EgPs with each cell line on a 96-well microplate for 72 h (37 °C, 5% CO_2_ in a humid atmosphere). The cell line proliferations were estimated by the MTT (3-(4,5-dimethylthiazol-2-yl)-2,5-diphenyltetrazolium bromide) assay (Sigma-Aldrich, St. Luis, MO, USA) [78]. Determinations were carried out in triplicate in independent experiments.

### 4.16. Determination of Cytokines with RAW 264.7 Cell Line

RAW 264.7 cells were cultured in the presence of different concentrations of polysaccharides (0–100 μg mL^−1^) in 96-well microplates (5 × 10^4^ cells well^−1^) in a total volume of 100 μL. Bacterial LPS (50 ng mL^−1^) was used as a positive control for the activation of macrophages and wells with treated and untreated medium, for 24 h. After this, the culture medium was taken from all the wells and tumor necrosis factor alpha (TNF-α) and interleukin-6 (IL-6) were quantified, using the ELISA Ready-SET-Go mouse TNF alpha kit and a mouse IL-6 Ready-SET-Go ELISA (Invitrogen, Thermo Fisher Scientific, Madrid, Spain). following the supplier’s instructions.

### 4.17. Determination of Cytokines with Human THP-1 Cell Line

Human THP-1 cells were seeded in 96-well microplates (5 × 10^4^ cells per well) and derived into macrophages after a 10 ng mL^−1^ PMA stimulation for 48 h. Then, wells were washed with fresh medium and exposed to different polysaccharide concentrations for a further 24 h. Human TNF-α present in the conditioned media was quantified by using a Human TNF alpha uncoated ELISA kit (Invitrogen, Thermo Fisher Scientific, Spain).

### 4.18. Proteomics Analysis

To investigate the effects of EgPs on protein expression levels in non-tumor cells, proteomics analysis of the HGF-1 cell line was carried out.

#### 4.18.1. Cell Treatment and Protein Extraction

HGF-1 cells were cultured in DMEM without fetal bovine serum and treated with 320 μg mL^−1^ of EgPs for 24 h, the highest concentration that produced no cytotoxic effect on cells was noted. After treatments, cells were washed with ice-cold phosphate-buffered saline and solubilized in 100 mM triethylammonium bicarbonate buffer (TEAB)-10% SDS containing Pierce universal nuclease (Thermo Scientific, MA). Then, cells were sonicated, centrifuged at 16,000× *g* for 10 min at 4 °C, and the supernatant was carefully separated.

#### 4.18.2. In-Solution Tryptic Digestion and 2-Plex Tandem Mass Tag (TMT) Labeling

The protein concentrations of samples were determined by the bicinchoninic acid protein assay and normalized to the same concentration (1 ug μL^−1^). For reduction and alkylation, 5 μL de 200 mM Tris (2-carboxyethyl) phosphine was added and incubated at 55 °C for 1 h. Proteins were alkylated with 60 mM iodoacetamide at room temperature for 30 min, protected from light. Samples were subjected to acetone precipitation to purify proteins by incubating with 6 volumes of ice-cold acetone at −20 °C for 4 h. Precipitated proteins were centrifuged at 8000× *g* for 10 min at 4 °C and the pellet redissolved in 100 μL of 50 mM TEAB (pH 8.5). Proteins were digested by trypsin (Pierce trypsin protease, Thermo Scientific, MA) at a ratio of 1:50 (trypsin:protein, *w*/*w*) by incubating overnight at 37 °C.

For quantitative proteomics, each digested protein sample was labelled with an appropriate TMT2plex Isobaric Label Reagent (Thermo Scientific, MA), according to the protocol supplied by the manufacturer. Briefly, the control and treated peptide samples were labelled with reagents TMT-126 and TMT-127, respectively. To quench the TMT reaction, 8 μL of 5% hydroxylamine was added to each sample and incubated for 15 min at room temperature. All the labelled peptides were combined in equal amounts and dried using a speed vacuum system.

#### 4.18.3. Liquid Chromatography High-Resolution Mass Spectrometry

The peptides were dissolved in 0.1% formic acid and analyzed using a Q-Exactive HF-X Hybrid Quadrupole-Orbitrap Mass Spectrometer (Thermo Scientific, MA, USA) connected to an Easy-nLC 1200 UHPLC (Thermo Scientific, MA, USA). Solvent A and solvent B were 0.1% formic acid in water and 0.1% formic acid in 80% acetonitrile, respectively. Peptide separation was performed using a trap column (Acclaim PepMap 100, 75 μm × 2 cm, C18, 3 μm, 100 A, Thermo Scientific, MA, USA) at a flow rate of 20 μL min^−1^ and eluted onto a 50 cm analytical column (PepMap RSLC C18, 2 μm, 100 A, 75 μm x 50 cm, Thermo Scientific) with a linear gradient of 2% to 20% solvent B (0.1% formic acid in 80% acetonitrile) for 240 min, followed by a 30 min gradient from 20% to 35% solvent B and, finally, to 95% solvent B for 15 min before re-equilibration to 2% solvent B at a constant flow rate of 300 nL min^−1^. The peptides were positively ionized using a nanospray ion source and mass spectrometry was conducted in top 20 data-dependent mode with the following settings: 2.2 kV ion spray voltage; 320 °C ion capillary temperature, 350–1500 mass scan range for full MS and 350–1500 scan range for MS/MS; MS resolution 120,000 for full MS and 30,000 for data-dependent MS/MS; collision energy 32 with high-energy collisional dissociation mode; MS/MS isolation window 0.7 *m*/*z*; and dynamic exclusion 20 s.

#### 4.18.4. Data Analysis

The peptides and proteins were identified and quantified using Proteome Discoverer^TM^ 2.4 (Thermo Scientific, MA, USA) platform. Sequest HT^®^ was utilized as the search engine [82], and the Swiss-Prot part of UniProt for *Homo sapiens* was used as the database, accessed on 29 September 2021. The search parameters were set as follows: two missed tryptic cleavage site was allowed; full MS and MS/MS tolerances were 10 ppm and 0.02 Da, respectively; fixed modification of carbamidomethylation on Cys, variable modification of oxidation on Met, and *N*-terminal acetylation on protein were specified; proteins required a minimum of two peptides. Peptide spectral matches and consecutive protein assignments were validated using the Percolator^®^ algorithm [83] by imposing a strict cut-off of 1% false discovery rate. To calculate the ratio of TMT-labelled proteins between treated and control cells, abundances were based on precursor intensities. Normalization was performed based on total peptide amount, and samples were scaled on all average (for every protein and peptide the average of all samples is 100).

The Search Tool for the Retrieval of Interacting Genes/Proteins (STRING) (https://www.string-db.org, accessed on 13 October 2021) was used to perform a protein-protein interaction networks functional enrichment analysis.

Three technical replicates of each sample were carried out.

### 4.19. Statistical Analysis

The statistical analyses were performed with the software Statistica 7.0. Figures were made using SigmaPlot versión 12.0, 2015 (Systat Software Inc., Chicago, IL, USA). All values were expressed as means ± standard deviation (SD). To evaluate the differences of each treatment with polysaccharides, a one-way ANOVA was conducted. Later on, when significant differences were found, a post-hoc Tukey HSD test was conducted to compare the means between each treatment. Differences were considered to be statistically significant when *p* < 0.05.

In proteomics analysis, the normalized and scaled relative abundance of every protein was expressed as mean ± standard deviation of four biological replicates. Protein ratios were directly calculated from the grouped protein abundances. Abundance ratio *p*-values were calculated by ANOVA, based on the abundance of individual proteins. Only proteins with ANOVA *p* < 0.05 and Log_2_ fold change > 0.3 were considered as significantly deregulated.

## 5. Conclusions

The present study concludes that EgPs have potential anticancer, immunomodulatory and antioxidant capacity, in addition to overexpressing five proteins that interact with each other and with TNF-α and IL-6. As this seaweed is a natural product, and its extracted polysaccharides do not present side effects, at least as far as this study has been able to determine, it should be considered a potential nutraceutical.

## Figures and Tables

**Figure 1 pharmaceuticals-15-01379-f001:**
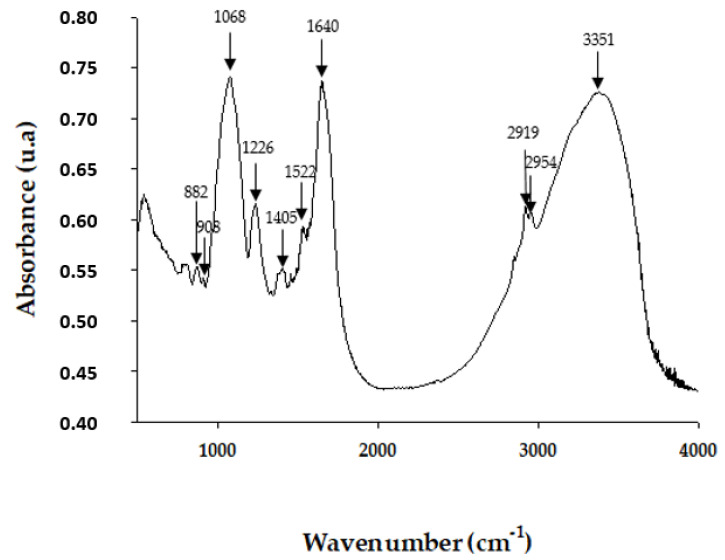
FT-IR spectra of the EgPs.

**Figure 2 pharmaceuticals-15-01379-f002:**
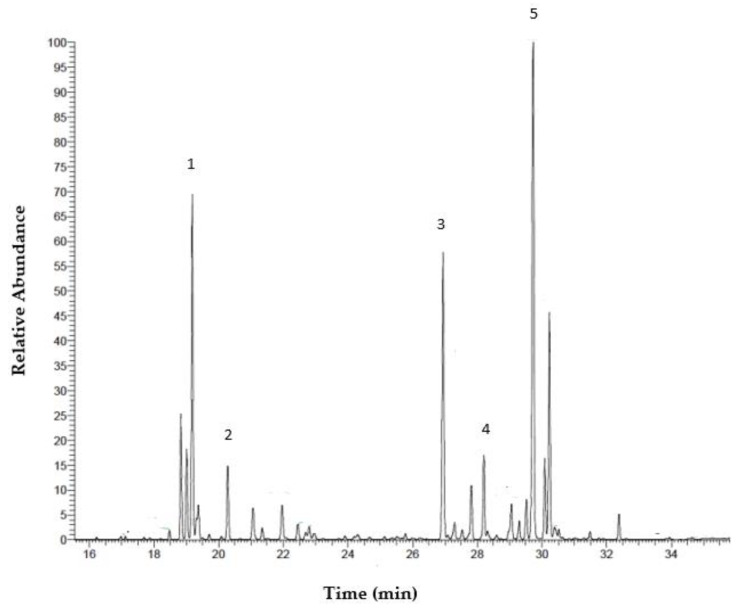
GC–MS of the EgPs. For peak description see Table 3.

**Figure 3 pharmaceuticals-15-01379-f003:**
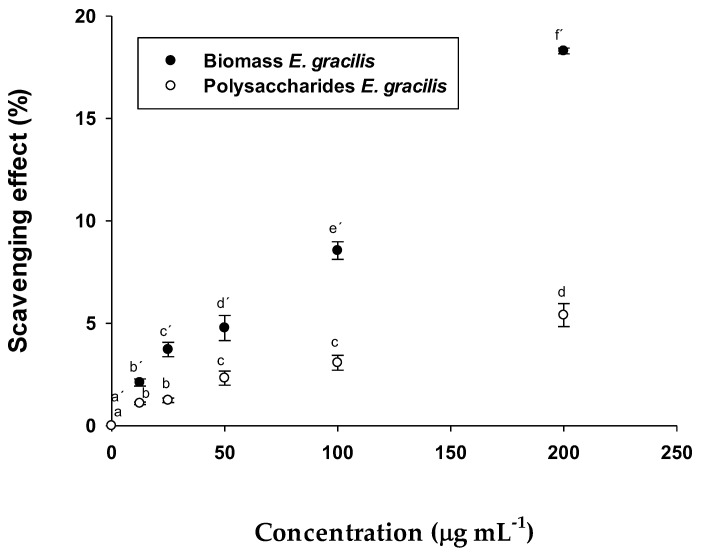
Scavenging effects % of the sample on ABTS radical. The data are the mean of 3 replicate measurements ± standard error. Similar letters indicate no significant differences (Tukey, *p* < 0.05) between the different concentrations.

**Figure 4 pharmaceuticals-15-01379-f004:**
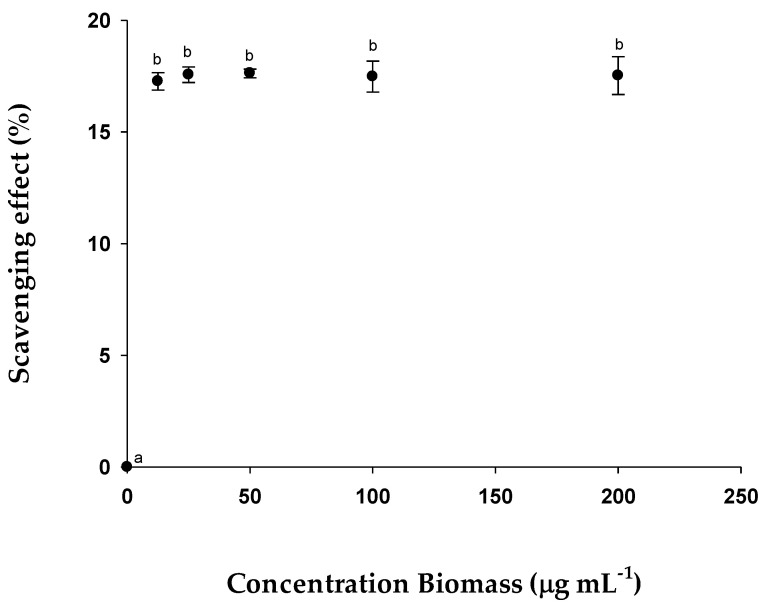
Scavenging effects % of the sample on DPPH radical. The data are the mean of 3 replicate measurements ± standard error. Similar letters (a or b) indicate no significant differences (Tukey, *p* < 0.05) between the different concentrations.

**Figure 5 pharmaceuticals-15-01379-f005:**
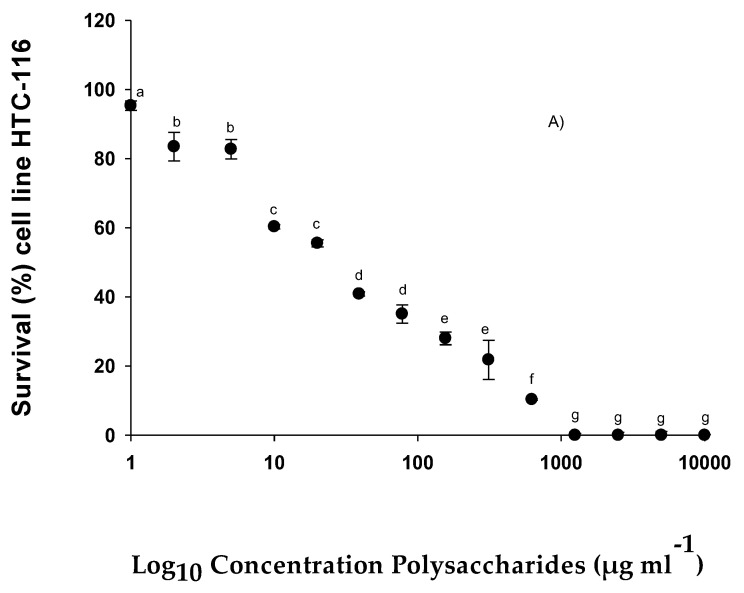
% Survival of different cell lines exposed to different concentrations of polysaccharides from *E. gracilis*: (**A**) % Survival of cell line HTC-116; (**B**) % Survival of cell line NCl-H460; (**C**) % Survival of cell line MCF-7; (**D**) % Survival of cell line U-937; (**E**) % Survival of cell line HL-60. Similar letters indicate that there are no significant differences (Tukey, *p* < 0.05) between the different concentrations.

**Figure 6 pharmaceuticals-15-01379-f006:**
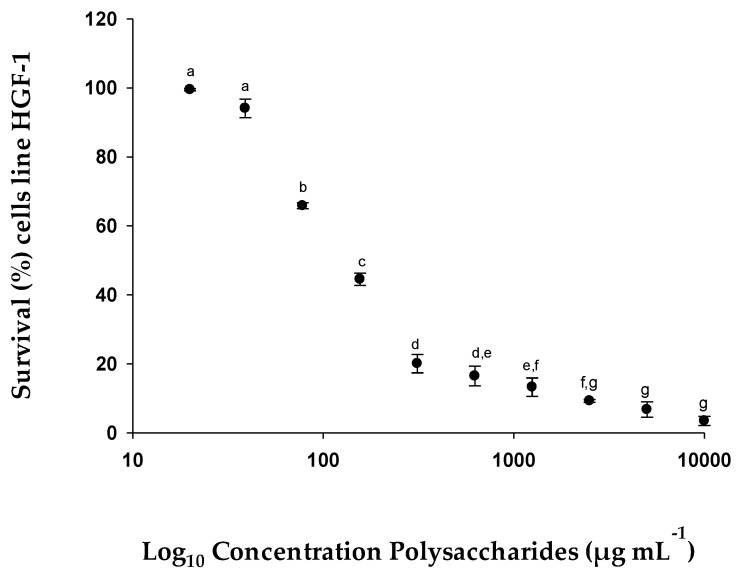
% Survival of cell line HGF-1 (human gingival fibroblasts), very proliferative, exposed to different concentrations of EgPs. Similar letters (a to g) indicate that there were no significant differences (Tukey, *p* < 0.05) between the different concentrations.

**Figure 7 pharmaceuticals-15-01379-f007:**
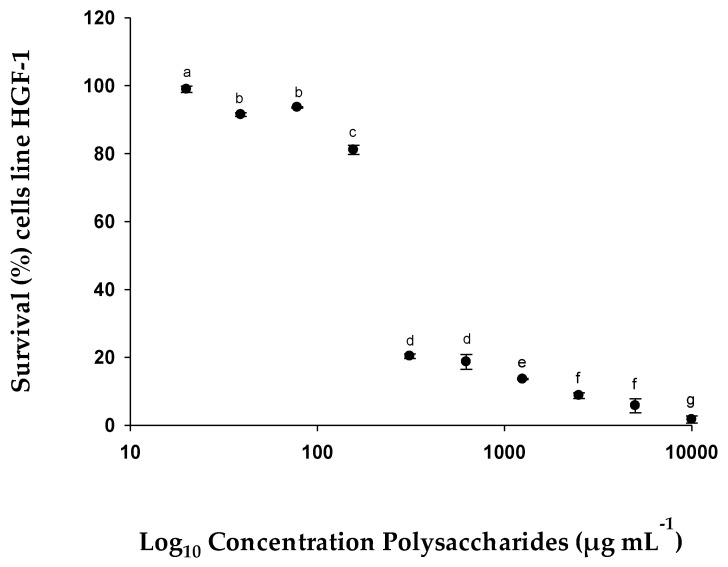
% Survival of cell line HGF-1 (human gingival fibroblasts), low proliferative, exposed to different concentrations of EgPs. Similar letters (a to g) indicate that there are no significant differences (Tukey, *p* < 0.05) between the different concentrations.

**Figure 8 pharmaceuticals-15-01379-f008:**
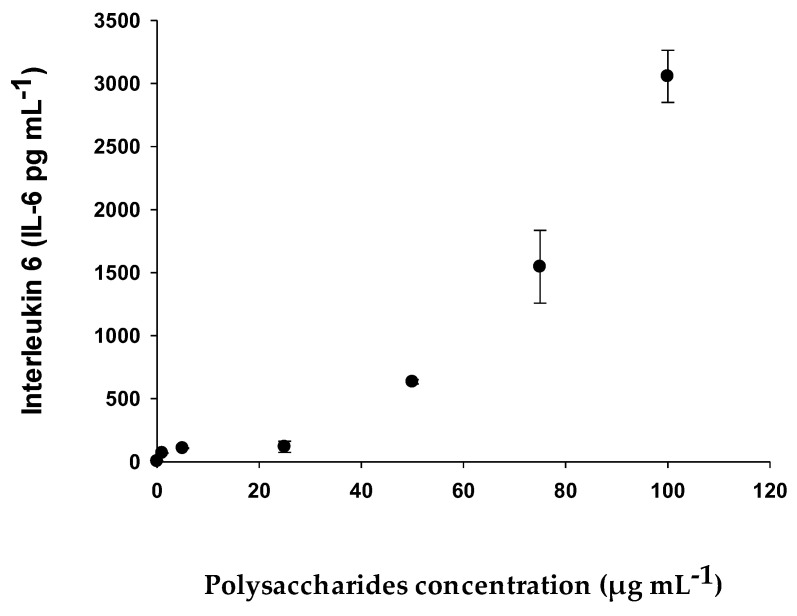
Production of Interleukin-6 by RAW 264.7 macrophages exposed to different concentrations of EgPs. Data points represent the average of three samples ± SD.

**Figure 9 pharmaceuticals-15-01379-f009:**
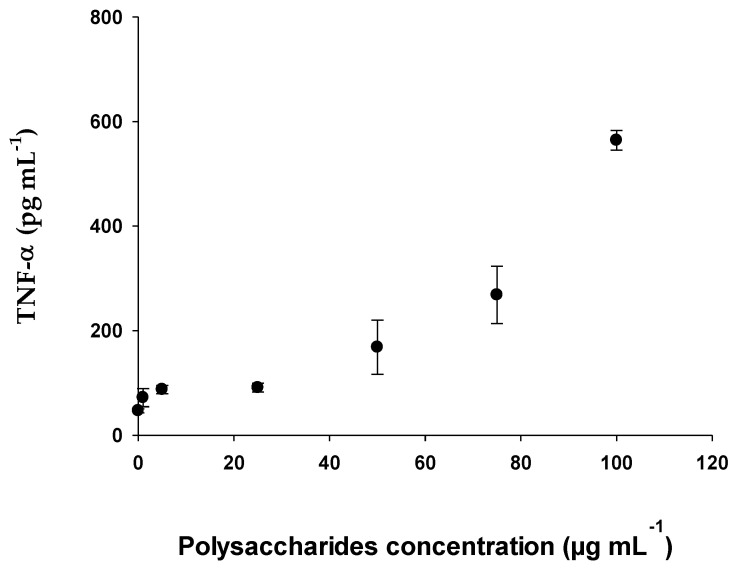
Concentration of Tumor Necrosis Factor-α by RAW 264.7 macrophages exposed to different concentrations of EgPs. Data points represent the average of three samples ± SD.

**Figure 10 pharmaceuticals-15-01379-f010:**
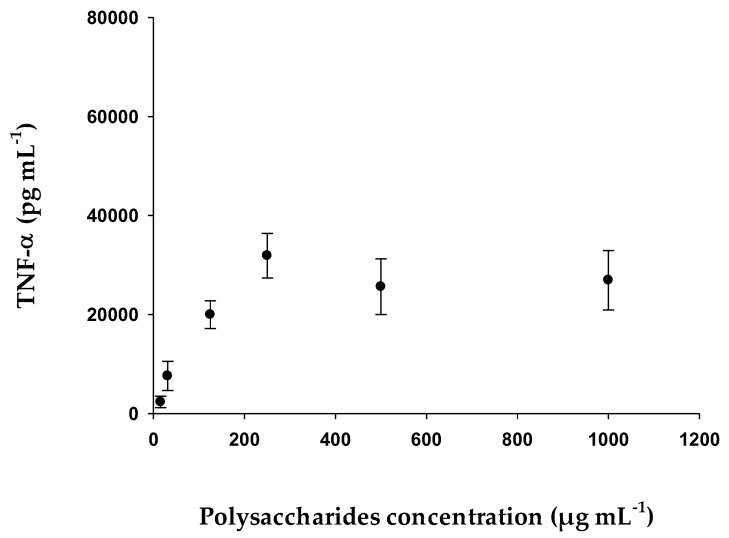
Concentration of Tumor Necrosis Factor-α by HPT-1 exposed to different concentrations of EgPs. Data are the mean SD of three independent experiments.

**Figure 11 pharmaceuticals-15-01379-f011:**
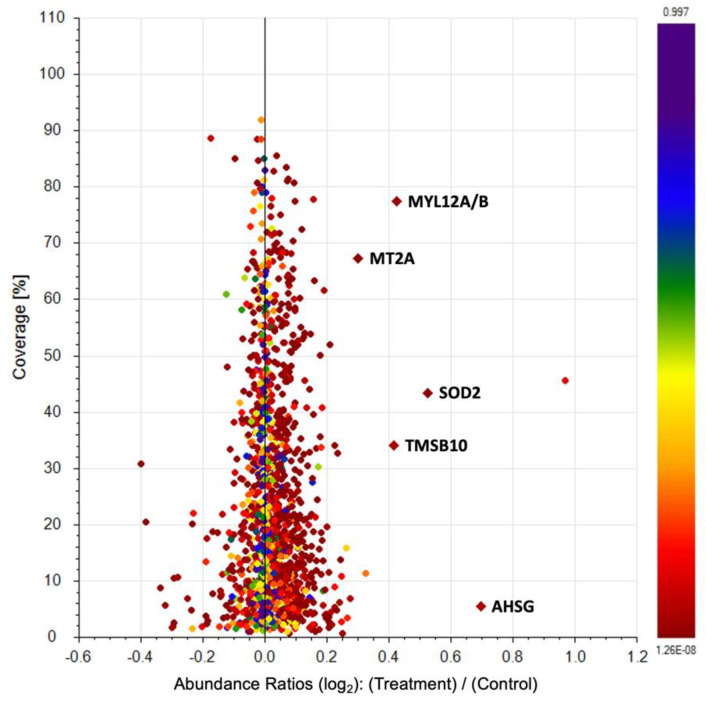
Scatter plots showing the percentage of protein sequence coverage vs. their Log_2_ fold change abundance value in HGF cells after 24 h treatment with EgPs. Proteins were ranked according to their *p*-value from red to blue. Proteins with *p*-value < 0.05 and Log_2_ fold change > 0.3 are drawn in diamonds showing their gene symbol.

**Figure 12 pharmaceuticals-15-01379-f012:**
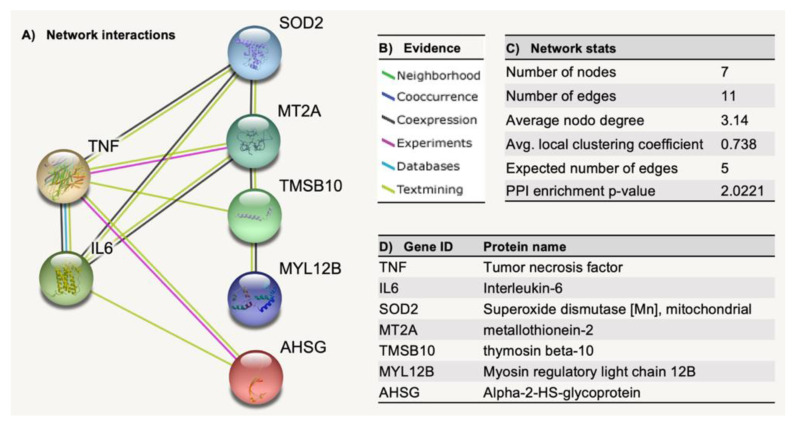
Protein–protein interaction networks functional enrichment analysis among the significantly upregulated proteins in human gingival fibroblast cells treated with EgPs and the key pro-inflammatory cytokines TNF-α and IL-6 using STRING, the Search Tool for the Retrieval of Interacting Genes/Proteins. Network nodes were proteins, and edges represented the predicted functional associations (**A**), drawn according to the type of evidence (**B**). Network statistics reporting data concerning the number of nodes and edges, the average node degree, the average local clustering coefficient, the expected number of edges, and the protein–protein interaction (PPI) enrichment *p*-value are shown in (**C**). Proteins, and their gene symbols, are listed in (**D**).

**Table 1 pharmaceuticals-15-01379-t001:** Total Carbon (C), Hydrogen (H), Nitrogen (N), and Sulphur (S) obtained in the biomass and extracted EgPs.

	Biomass (%)	Extracted Polysaccharides (%)
Carbon	44.59 ± 0.97	1.24 ± 0.82
Hydrogen	5.57 ± 0.41	0.20 ± 0.09
Nitrogen	10.54 ± 1.02	0.24 ± 0.07
Sulphur	0.03 ± 0.003	0.16 ± 0.06

**Table 2 pharmaceuticals-15-01379-t002:** Content of proteins, carbohydrates, lipids, inorganic compounds, and moisture in biomass from *E. gracilis* (% of DW).

Proteins	52.59± 0.69%
Carbohydrates	7.98 ± 1.23%
Lipids	11.30 ± 0.72%
Inorganic compounds	11.64 ± 1.38 %
Moisture	16.49 ± 1.98%

**Table 3 pharmaceuticals-15-01379-t003:** Content of monosaccharides in EgPs.

Header	Monosaccharide	Retention Time (min)	Peak Area	% Mass
1	Ribose (Rib)	19.19	122,217,046	19.58
2	Fucose (Fuc)	20.28	29,872,029	4.79
3	Mannose (Mann)	26.94	113,488,499	18.18
4	Galactose (Gal)	28.20	377,111,564	6.04
5	Glucose (Glc)	29.73	21,748,410	34.83

**Table 4 pharmaceuticals-15-01379-t004:** Significantly overexpressed proteins in HGF cells after treatment with EgPs.

UniProt Accession	Gene Symbol	Description	Sum PEP Score *	Abundance Ratio: Treatment/Control	Abundance Ratio *p*-Value
O14950	MYL12B; MYL12A	Myosin regulatory light chain 12A/B	81.34	1.34	2.96 × 10^−2^
P04179	SOD2	Superoxide dismutase [Mn], mitochondrial	30.42	1.44	2.48 × 10^−6^
P02795	MT2A	Metallothionein-2	28.35	1.23	1.90 × 10^−44^
P02765	AHSG	Alpha-2-HS-glycoprotein	16.59	1.62	4.83 × 10^−22^
P63313	TMSB10	Thymosin beta-10	13.29	1.33	3.32 × 10^−22^

* The sum PEP score corresponds to the score calculated based on the posterior error probability (PEP) values of the peptide spectrum matches (PSM). The PEP indicates the probability that an observed PSM was a random event. The Sum PEP score was calculated as the negative logarithms of the PEP values of the connected PSM.

**Table 5 pharmaceuticals-15-01379-t005:** Network interaction data.

Node 1	Node 2	Score
AHSG	IL6	0.518
AHSG	TNF	0.474
IL6	AHSG	0.518
IL6	MT2A	0.344
IL6	SOD2	0.594
IL6	TNF	0.994
MT2A	IL6	0.344
MT2A	SOD2	0.336
MT2A	TMSB10	0.173
MT2A	TNF	0.281
MYL12B	TMSB10	0.201
SOD2	IL6	0.594
SOD2	MT2A	0.336
SOD2	TNF	0.619
TMSB10	MT2A	0.173
TMSB10	MYL12B	0.201
TMSB10	TNF	0.181
TNF	AHSG	0.474
TNF	IL6	0.994
TNF	MT2A	0.281
TNF	SOD2	0.619
TNF	TMSB10	0.181

## Data Availability

Data is contained within the article.

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
