# Peer review of "Antioxidant, Immunomodulatory and Potential Anticancer Capacity of Polysaccharides (Glucans) from Euglena gracilis G.A. Klebs"

_pharmaceuticals, 2022, doi:10.3390/ph15111379_

Round 1
Reviewer 1 Report
Authors have reported the antioxidant, immunomodulatory and potential anticancer capacity of polysaccharides (glucans) from Euglena gracilis G.A. Klebs the work is good but following revision is needed.
1. In the abstrct authors should mention whether the compounds was non toxic to normal cell line.
2. Why was NMR not done for compond please give reasons ?
3. For the antioxidant activity, anticancer and other activity some recent references is needed this may help https://doi.org/10.1038/s41598-018-38214-x; https://doi.org/10.5012/jkcs.2021.65.2.106
4.What was the difference in mtt assay for cancer cell line and normal cell lines.
5. What was different standard used for the activities it comparasion
6. How was the derivatized compounds charaterized ?
7. Conclusion must rewritten.
8. Some of result needs to be rechecked.
9. English gammar and reference need to be checked and formatted according to journal format.
Reviewer 2 Report
Extensive editing of English language and style required.
Observations:
Introduction
- Please check English language (ex pag 2: With over 30,000 species of microalgae, only about 100 have been studied and ten are 51 commercially relevant. Its ability to be grown on a large scale…)
- Rephormulate, please: The sulfated essential amino, sulfated polyunsaturated fatty acids [11], β-carotenes [8], and paramylon, a high molecular weight, unbranched β-(1,3)-D-glucan reserve polysaccharide with immunomodulatory properties [12–14], have also been reported in the specie.
- This research allowed an in depth study of EgPs and finding metabolites of interest as a source of bioinspiration. What for?
Results
- Total Carbon (TC), Total Nitrogen (TN), Total Hydrogen (TH) and Total Sulfur(TS) were evaluated for biomass and EgPs. Please comment the obtained results!
- Please improve the quality of figures!
- Biological assessment: The authors present antioxidant activity (DPPH method) only for biomass. What about EgPs? Other biological properties (Cell viability of lines HTC-116, MCF-7, U-937, HL-60 and NCl-H460, cytokines (IL-6 and TNF-α) with RAW 264.7 cell line… are presented for polysaccharides? There is an explanation?
- These values indicate that the EgPs are more active in proliferating cells, thus, cancer cells. Please re-phormulate!
Discussion
- According to Meng et al. (2014) [33], the FTIR method is a validated spectroscopic method, which characterizes the polysaccharides of algae, determines the variations of other primary metabolites and evaluates the physiology of the microalgae. I did not find something about this in your article!
Conclusions
Please re-phormulate, according to your data!
Round 2
Reviewer 1 Report
The present study was carried out to determine the bioactivity of polysaccharides extracted from Euglena gracilis (EgPs). The following should be done
The compounds were characterized by FT-IR and GC-MS but charactrization with HPLC or HPTLC is also needed.
Reviewer 2 Report
-